:ᐲ: PLOS ONE

# Effects of the killer immunoglobulin–like receptor (*KIR*) polymorphisms on HIV acquisition: A meta-analysis

Suwit Chaisri[1,2], Noel Pabalan[1]*, Sompong Tabunhan[1], Phuntila Tharabenjasin[1], Nipaporn Sankuntaw[1], Chanvit Leelayuwat[2,3]

**1** Chulabhorn International College of Medicine, Thammasat University, Pathum Thani, Thailand, **2** The Centre for Research and Development of Medical Diagnostic Laboratories (CMDL), Faculty of Associated Medical Sciences, Khon Kaen University, Khon Kaen, Thailand, **3** Department of Clinical Immunology and Transfusion Sciences, Faculty of Associated Medical Sciences, Khon Kaen University, Khon Kaen, Thailand

* noelpabalan@mail.com

**Data Availability Statement:** All relevant data are within the manuscript and its Supporting Information files.

## Abstract

### Background

Genetic involvement of *Killer Immunoglobulin-like Receptor (KIR)* polymorphisms and Human Immunodeficiency Virus (HIV)-exposed seronegative (HESN) compared to HIV-infected (HIVI) individuals has been reported. However, inconsistency of the outcomes reduces precision of the estimates. A meta-analysis was applied to obtain more precise estimates of association.

### Methods

A multi-database literature search yielded thirteen case-control studies. Risks were expressed as odds ratios (ORs) and 95% confidence intervals (CIs) with significance set at a two-tailed P-value of $\leq$ 0.05. We used two levels of analyses: (1) gene content that included 13 *KIR* polymorphisms (*2DL1-3*, *2DL5A*, *2DL5B*, *2DS1-3*, *2DS4F*, *2DS4D*, *2DS5*, *3DL1* and *3DS1*); and (2) *3DL1/S1* genotypes. Subgroup analysis was ethnicity-based (Caucasians, Asians and Africans). Outlier treatment was applied to heterogeneous effects which dichotomized the outcomes into pre-outlier (PRO) and post-outlier (PSO). Multiple comparisons were addressed with the Bonferroni correction.

### Results

We generated 52 and 18 comparisons from gene content and genotype analyses, respectively. Of the 70 comparisons, 13 yielded significant outcomes, two (indicating reduced risk) of which survived the Bonferroni correction (P$^c$). These protective effects pointed to the Caucasian subgroup in *2DL3* (OR 0.19, 95% CI 0.09, 0.40, P$^c$ < 10$^{-3}$) and *3DS1S1* (OR 0.37, 95% CI 0.24, 0.56, P$^c$ < 10$^{-3}$). These two PSO outcomes yielded effects of increased magnitude and precision, as well as raised significance and deemed robust by sensitivity analysis. Of the two, the *2DL3* effect was improved with a test of interaction (P$^{c\ interaction}$ < 10$^{-4}$).

**Funding:** All the funding or sources of support received during this study was from the Thailand Research Fund and the Commission on Higher Education under the Grant for New Researcher (No.MRG 6180172) directed to the Principal Investigator, Dr. Suwit Chaisri. There was no additional external funding received for this study.

**Competing interests:** The authors have declared that no competing interests exist.

**Abbreviations:** [R], Reference number; CB, Clark-Baudouin; CI, Confidence interval; HESN, Human Immunodeficiency Virus-exposed seronegative; HIV, Human Immunodeficiency Virus; HIVI, HIV-infected; HLA, Human Leukocyte Antigen; $I^2$, Heterogeneity test; IQR, Interquartile range; K, Number of studies; KIR, Killer Immunoglobulin-like receptor; maf, minor allele frequency; n, Sample size or number of individuals; N, Total number of studies; NK, Natural Killer cells; OR, Odds ratio; $P^a$, P-value for test of association (pre-Bonferroni; $P^b$, P-value for test of heterogeneity; $P^c$, Bonferroni-corrected P-value for association; $P^{ci}$, Bonferroni-corrected P-value for interaction; PRISMA, Preferred Reporting Items for Systematic Reviews and Meta-Analyses; PRO, Pre-outlier; PSO, Post-outlier; SD, Standard deviation.

## Conclusion

Multiple meta-analytical treatments presented strong evidence of the protective effect (up to 81%) of the *KIR* polymorphisms (*2DL3* and *3DS1S1*) among Caucasians. The Asian and African outcomes were inconclusive due to the low number of studies.

## Introduction

Natural Killer (NK) cells are key effectors of innate immunity in response to virus-infected and transformed cells [1, 2]. NK cell functions are regulated by the balance of signal transduction through their activating and inhibitory receptors. Effector functions of NK cells include direct cytotoxic activity and cytokine release [3]. Killer Immunoglobulin-like receptors (KIRs) are highly polymorphic glycoproteins expressed on NK cells. Genetic diversity of *KIRs* includes variations in gene content and copy number as well as allelic polymorphisms [4–8]. *KIR* members include 15 functional genes (*2DL1-4*, *2DL5A*, *2DL5B*, *2DS1-5*, *3DL1-3* and *3DS1*), and 2 pseudogenes (*2DP1*, *3DP1*). KIR ligands are human leukocyte antigen (HLA)-class I molecules that are expressed in all nucleated cells. The interactions between KIR and HLA class I molecules regulate NK cell function. To date, impact of *KIR* diversity has been investigated in several human diseases and conditions that include infection, autoimmunity, inflammatory disorders, hematopoietic stem transplantation and reproduction [9]. Recent studies have shown that *KIR* polymorphisms are associated with susceptibility to Human Immunodeficiency Virus (HIV)-1 infection and HIV disease progression [10–12]. In addition, *3DL1/S1* locus is unusual in that it shows allelic polymorphisms encoding inhibitory (*3DL1*) or activating (*3DS1*) receptors [13, 14]. These *3DL1/S1* functions have been reported as protecting against HIV-infection and progression [15–18]. Moreover, increasing numbers of association studies of *3DL1/S1* and HIV acquisition have compared HIV-infected (HIVI) and HIV-exposed seronegative (HESN) individuals. HESN individuals are those who resist HIV-infection despite repeated exposure to the virus. HESN individuals were found to have enriched *3DL1/S1* genotypes [19]. The mechanism by which HESN individuals are naturally protected renders this group as more suitable than healthy controls [19, 20]. Therefore, the resistance of such individuals to HIV has been the focus of interest in identifying the mechanisms of natural protection. For HESN individuals with *3DS1* and/or *3DL1*, it has been proposed that both *KIR* polymorphisms are required for increased NK cell activity in the killing of HIV-infected cells [21]. However, not all studies agree with KIR's role in HIV infection [22], rendering inconsistency to the cumulative outcomes of the reported studies. Their conclusions may have been limited by inadequate statistical power because of small sample sizes and lack of proportional controls. Given these inconsistencies, we perform a meta-analysis to obtain better estimates of precision and statistical power to help establish associations of the *KIR* polymorphisms with HIV acquisition.

## Materials and methods

### Search strategy

Three databases (PubMed, Google Scholar and Science Direct) were searched for association studies as of November 28, 2018. The terms used were "*Killer Immunoglobulin-like Receptor*", '*KIR*", "*HIV*", "*Human Immunodeficiency Virus*", "*HESN*" "*HIV-exposed seronegative*" as

medical subject headings and text, without language restrictions. References cited in the retrieved articles were screened manually to identify additional eligible studies.

## Inclusion and exclusion criteria

SC and NP independently decided on which articles were to be included. This was then discussed in order to reach an agreement; otherwise, NS adjudicated so that consensus was obtained. Inclusion criteria included the following: (1) articles evaluating associations between *KIR* polymorphisms and risk for HIV acquisition; (2) the studies have a case–control study design; (3) HIVI cases; (4) controls were HESN, tested with HIV enzyme immunoassay or reverse transcriptase-polymerase chain reaction for at least 18 months; (5) sufficient genotype or allele frequency data to allow calculation of odds ratios (ORs) and 95% confidence intervals (CIs). Excluded articles were those that: (1) evaluated associations between *KIR* polymorphisms and HIV progression; (2) had no controls or with healthy controls; (3) unconfirmed HIV infection; (4); were reviews; (5) had duplicate data; (6) had incomplete or absent genotype data.

## Data extraction

Two investigators (SC and NP) independently extracted data and reached a consensus on all the items, adjudicated by a third investigator (NS). The following information was obtained from each publication: (i) first author's name; (ii) published year; (iii) country of origin; (iv) ethnicity; (v) total sample sizes; (vi) number of HIVI and HESN; (vi) genotyping platform; (vii) *KIR* gene content polymorphisms: (viii) *KIR3DL1/S1* genotypes and minor allele frequencies. In attempts to fill missing information, we contacted the primary-study authors. None of the included studies mentioned the influence of environment, nor were data provided.

## Quality of the studies

SC and NP assessed the methodological quality of the included studies. The Clark-Baudouin (CB) scale was used for this purpose [23] because it focuses on statistical (P-values, power and corrections for multiplicity) and genetic (genotyping methods) features of the included studies. CB scores range from 0 (worst) to 10 (best) where quality is rated as low (< 5), moderate (5–6) and high (7–10).

## Data synthesis

Risks of HIV acquisition (using raw data for frequencies) were estimated for each study wherein ORs were calculated for the 13 KIR genes (*2DL1-3*, *2DL5A*, *2DL5B*, *2DS1-3*, *2DS4D*, *2DS4F*, *2DS5*, *3DL1* and *3DS1*) and the *3DL1/S1* genotypes. The framework and pseudogenes were excluded for analysis (*2DL4*, *3DL2*, *3DL3*, *2DP1* and *3DP1*) because of their presence in all haplotypes. Gene content analysis (presence/absence) was based on the frequency data of HIVI and HESN. Use of HESN as controls precluded testing for Hardy-Weinberg Equilibrium. The combination of gene content variation and genotype distribution precluded the use of standard genetic modeling, but allowed application of the allele genotype model. Subgrouping was ethnicity-based (Asian, Caucasian and African). Heterogeneity between studies was estimated using the chi-square based Q-test [24], and quantified with the $I^2$ statistic which measures degree of inconsistency between studies [25]. An $I^2 \geq 50\%$ with $P \leq 0.10$ indicated the presence of heterogeneity, which prompted use of the random-effects model [26], otherwise the fixed- effects model was used [27]. Sources (outlying studies) of heterogeneity were detected with the Galbraith plot [28]. Outlier treatment consisted of eliminating sources of

heterogeneity followed by reanalysis. Differential outcomes between the ethnicities (Asians, Caucasians or Africans) warranted tests of interaction [29]. Threshold for significance was set at $P \leq 0.05$ (two-sided) except in estimations of heterogeneity [30]. Multiple comparisons were Bonferroni-corrected. Sensitivity analysis, which involves omitting one study at a time followed by recalculation, was used to test for robustness of the summary effects. Publication bias assessment was contingent on two conditions: i) statistically significant associations and ii) comparisons with $\geq 10$ studies; less than this number reduces sensitivity of the qualitative and quantitative tests [31]. Distribution of continuous data was assessed with the Shapiro-Wilk (SW) test [32]. Normal distribution warranted the use of mean ± standard deviation (SD) and the parametric approach. Otherwise, non-normal data distribution was descriptively expressed as median and interquartile range (IQR), with an inferential non-parametric approach. Data were analyzed using Review Manager 5.3 (Cochrane Collaboration, Oxford, England), SIGMASTAT 2.03, SIGMAPLOT 11.0 (Systat Software, San Jose, CA).

# Results

## Characteristics of the included studies

Fig 1 outlines the study selection process in a flowchart following PRISMA (Preferred Reporting Items for Systematic Reviews and Meta-Analyses) guidelines [33]. Initial search yielded a total of 325 citations; title and abstract screenings reduced this number to 51. Thirty four articles were excluded for not meeting our inclusion criteria; in addition, 4 articles/studies had absent or incomplete data (S1 List).

These series of exclusions resulted in 13 articles (studies) included in the meta-analysis [34–46]. Of the 13, three were included in the gene content analysis [35, 36, 39], five in the genotype analysis [34, 40–43] and five included both analyses [37, 38, 44–46]. Table 1 identifies which (Yes) articles cover gene content and genotype analyses. A total of 2,157 HIVI cases and 1,235 HESN controls were included in the meta-analysis (S1 and S2 Tables). S1 details the *KIR* polymorphisms for the gene content analysis and S2 outlines the *KIR3DL1/S1* genotypes (HIVI and HESN) for the genotype analysis. The number of articles included seven with Caucasian subjects (1,313 cases /485 controls)[40–44, 46, 47]; two Asians (256 cases /151 controls) [38, 39] and four Africans (588 cases /599 controls) [34–37]. Non-normal distribution of the CB scores (SW, $P = 0.04$) indicated high methodological quality of the included articles (median: 7, IQR: 6–8). S1 and S2 Tables show the quantitative traits of the included studies. Total sample sizes ranged from 41 to 577. Statistical power of the individual studies was low, but high at the aggregate level (99.9%) at $\alpha = 0.01$ and OR of 1.5 (G*Power program: http://www.psycho.uni-duesseldorf.de/aap/-projects/gpower). A detailed description of our study is summarized for PRISMA (S3 Table) and for genetic association studies (S4 Table).

## Overall comparisons

**Gene content analysis.** Table 2 shows eight significant outcomes, the $P^a$ values of which ranged from high ($< 10^{-5}$) to marginal (0.05). Risks were increased in five and decreased in three outcomes. On account of two polymorphisms (*2DS4F* and *3DS1*), risks in the overall analysis were increased (OR 1.62, 95% CI 1.10, 2.37) and decreased (OR 0.76, 95% CI 0.57, 1.00), respectively. Subgroup-wise, Caucasians were susceptible on account of *2DL2* (OR 1.36, 95% CI 1.00, 1.84) and *2DS1* (OR 1.71, 95% CI 1.15, 2.53). Contrastingly, this subgroup was protected because of *2DL1* (OR 0.20, 95% CI 0.05, 0.79) and *2DL3* (OR 0.29, 95% CI 0.11, 0.75). Risks were increased for Asians (*2DL5B*: OR 2.80, 95% CI 1.17, 6.67) and Africans (*2DS4F*: OR 2.01, 95% CI 2.01, 3.18). Of note, only the *2DL3* polymorphism in Caucasians

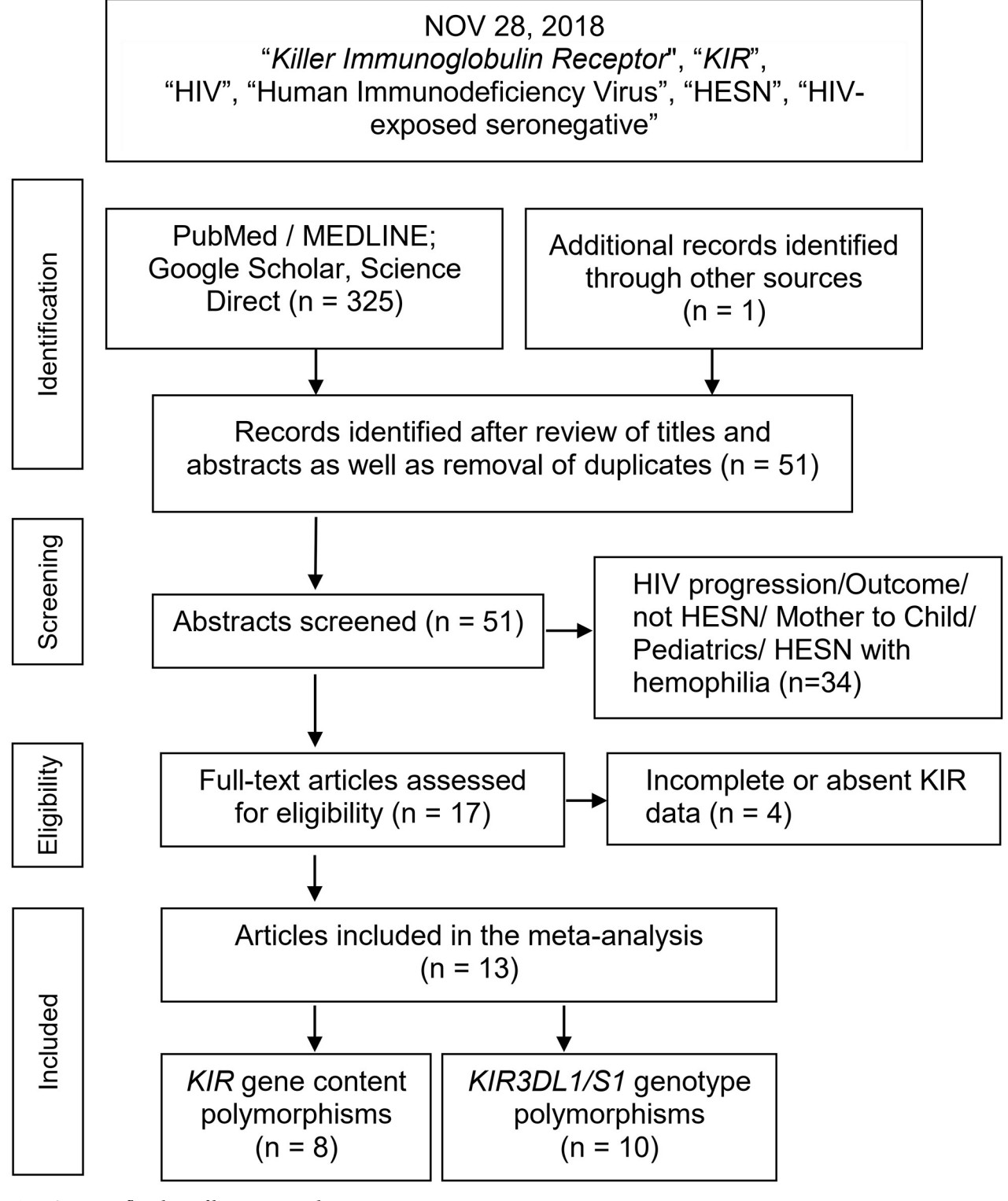

**Fig 1. Summary flowchart of literature search.**

**Table 1. Characteristics of the studies in the *KIR* polymorphisms and its associations with HIV acquisition.**

| K | First author | Year | Country | Ethnic Group | *KIR* gene content polymorphisms | *3DL1/S1* genotype polymorphisms | [R] |
|---|---|---|---|---|---|---|---|
| 1 | Jennes | 2006 | Tanzania | African | No | Yes | [34] |
| 2 | Merino | 2011 | Zambia | African | Yes | No | [35] |
| 3 | Koehler | 2013 | Tanzania | African | Yes | No | [36] |
| 4 | Naranbhai | 2016 | South Africa | African | Yes | Yes | [37] |
| 5 | Chavan | 2014 | India | Asian | Yes | Yes | [38] |
| 6 | Mori | 2015 | Thailand | Asian | Yes | No | [39] |
| 7 | Boulet | 2008 | Canada | Caucasian | No | Yes | [40] |
| 8 | Guerin | 2011 | Italy | Caucasian | No | Yes | [41] |
| 9 | Habegger | 2013 | Argentina | Caucasian | No | Yes | [42] |
| 10 | Tallon | 2014 | Canada | Caucasian | No | Yes | [43] |
| 11 | Zwolinska | 2016 | Poland | Caucasian | Yes | Yes | [44] |
| 12 | Jackson | 2017 | Canada | Caucasian | Yes | Yes | [45] |
| 13 | Rallon | 2017 | Spain | Caucasian | Yes | Yes | [46] |

K; number designation of each article, [R]; reference number

(OR 0.19, 95% CI 0.09, 0.40, $P^a < 10^{-5}$) survived the Bonferroni correction ($P^c < 10^{-3}$) which centralizes this finding for gene content analysis.

**Genotype analysis of *3DL1/S1*.** Table 3 shows three significant outcomes ($P^a = 0.01–0.04$) in PRO, none of which survived the Bonferroni-correction ($P^c = 0.7$ to $> 1$) except *3DS1S1* in PSO ($P^c < 10^{-3}$) and this represents the core finding in our genotype analysis (Table 4). Figs 2–4 summarize the mechanism of outlier treatment of this polymorphism. Fig 2 shows in Caucasians, that the PRO reduced risk effect (OR 0.45, 95% CI 0.24, 0.84, $P^a = 0.01$) was heterogeneous ($P^b < 0.06$, $I^2 = 50\%$). The source of this heterogeneity [44] is shown in Fig 3. Fig 4 shows the PSO outcome (OR 0.37, 95% CI 0.24, 0.56, $P^a < 10^{-5}$) of intensified significance and reduced heterogeneity ($P^b = 0.38$, $I^2 = 5\%$).

**Tests of interaction.** S5 Table shows that of the 10 comparisons subjected to these tests, only the Caucasian effect in *2DL3* (OR 0.19, $P^a < 10^{-5}$) compared with that of the African effect (OR 1.23, $P^a = 0.23$) resulted in significant interaction ($P^{ci} < 10^{-4}$) suggesting improved association. Extent of the significant Caucasian effect is thus placed in context when compared with its non-significant African counterpart.

**Sensitivity analysis.** Table 5 shows all significant outcomes in the overall and subgroup analyses were unaffected by sensitivity treatment except the *2DL2*, *2DS1* and *3DS1* (gene content analysis) and *3DL1/S1* in PRO Caucasians (genotype analysis).

**Publication bias.** Two outcomes (*3DL1L1* and *3DS1S1*) in our meta-analysis had 10 studies which we subjected to the funnel plot analysis and tests for publication bias. Operating data (ORs) for *3DL1L1* and *3DS1S1* were respectively non-normal (SW: P < 0.001) and normal (SW: P = 0.053). Neither the *3DL1L1* (Begg Mazumdar: Kendall's tau = 0.07, P = 0.79) and *3DS1S1* (Egger's test: intercept: -0.40, P = 0.77) outcomes nor the funnel plot show evidence of publication bias (Fig 5).

## Discussion

### Summary of findings

Lack of evidence (mainly low number of studies) precluded conclusions about Asians and Africans. Our main findings are thus confined to Caucasians, who are afforded protection by two *KIR* polymorphisms (*2DL3* and *3DS1S1*) on account of a number of meta-analysis

**Table 2. Associations of KIR gene content polymorphisms with HIV acquisition.**

| *KIR* | Ethnicity | K | HIVI (n/N) | HESN (n/N) | Test of association | | | | | Test of heterogeneity | | AM |
|---|---|---|---|---|---|---|---|---|---|---|---|---|
| | | | | | OR | 95% CI | Risk | P[a] | P[c] | P[b] | I² (%) | |
| *Inhibitory KIR gene* | | | | | | | | | | | | |
| *2DL1* | All | 4 | 829/869 | 573/588 | 0.62 | 0.32, 1.22 | Decreased | 0.17 | >1 | 0.63 | 0 | F |
| | Caucasians | 3 | 580/643 | 238/244 | **0.20** | **0.05, 0.79** | **Decreased** | **0.02** | >1 | 0.11 | 54 | F |
| | Asians | 2 | 243/256 | 139/151 | 1.20 | 0.52, 2.74 | Increased | 0.67 | >1 | 0.15 | 52 | F |
| | Africans | 2 | 392/394 | 473/481 | 2.51 | 0.47, 13.34 | Increased | 0.28 | >1 | 0.28 | 15 | F |
| *2DL2* | All | 8 | 770/1,385 | 588/1,050 | 1.09 | 0.91, 1.29 | Increased | 0.35 | >1 | 0.48 | 0 | F |
| | Caucasians | 3 | 354/643 | 115/244 | **1.36** | **1.00, 1.84** | **Increased** | **0.05** | >1 | 0.38 | 0 | F |
| | Asians | 2 | 115/256 | 80/151 | 0.88 | 0.57, 1.36 | Decreased | 0.56 | >1 | 1.00 | 0 | F |
| | Africans | 3 | 301/486 | 393/655 | 1.00 | 0.79, 1.29 | Null | 0.97 | >1 | 0.55 | 0 | F |
| *2DL3* | All | 5 | 560/656 | 678/799 | 1.11 | 0.82, 1.49 | Increased | 0.50 | >1 | 0.76 | 0 | F |
| | Caucasians | 3 | 518/643 | 226/244 | **0.29** | **0.11, 0.75** | **Decreased** | **0.01** | 0.70 | 0.06 | 65 | R |
| | Caucasians* | 2 | 409/520 | 137/147 | **0.19** | **0.09, 0.40** | **Decreased** | **$< 10^{-5}$** | **$< 10^{-3}$** | 0.46 | 0 | F |
| | Asians | 2 | 238/256 | 128/151 | 2.02 | 0.32, 12.96 | Increased | 0.46 | >1 | 0.01 | 85 | R |
| | Africans | 3 | 417/486 | 553/655 | 1.23 | 0.88, 1.73 | Increased | 0.23 | >1 | 0.97 | 0 | F |
| *2DL5A* | All | 3 | 194/718 | 111/478 | 0.73 | 0.53, 1.01 | Decreased | 0.06 | >1 | 0.92 | 0 | F |
| | Caucasian | 1 | 147/431 | 42/105 | 0.78 | 0.50, 1.20 | Decreased | 0.26 | >1 | NA | NA | NA |
| | Africans | 1 | 19/240 | 37/326 | 0.67 | 0.38, 1.20 | Decreased | 0.18 | >1 | NA | NA | NA |
| *2DL5B* | All* | 3 | 268/718 | 228/478 | 1.14 | 0.68, 1.91 | Increased | 0.62 | >1 | 0.05 | 67 | R |
| | All | 2 | 233/671 | 204/431 | 0.91 | 0.69, 1.20 | Decreased | 0.51 | >1 | 0.67 | 0 | F |
| | Asians | 1 | 35/47 | 24/47 | **2.80** | **1.17, 6.67** | **Increased** | **0.02** | >1 | NA | NA | NA |
| | Caucasians | 1 | 108/431 | 30/105 | 0.84 | 0.52, 1.35 | Decreased | 0.46 | >1 | NA | NA | NA |
| | Africans | 1 | 125/240 | 174/326 | 0.95 | 0.68, 1.33 | Decreased | 0.76 | >1 | NA | NA | NA |
| *3DL1* | All | 8 | 1,335/1,390 | 1,016/1,050 | 1.03 | 0.64, 1.64 | Null | 0.91 | >1 | 0.28 | 20 | F |
| | Caucasians | 3 | 607/643 | 229/244 | 0.85 | 0.29, 2.44 | Decreased | 0.76 | >1 | 0.15 | 47 | F |
| | Asians | 2 | 240/256 | 137/151 | 1.17 | 0.54, 2.51 | Increased | 0.69 | >1 | 0.20 | 38 | F |
| | Africans | 3 | 488/491 | 650/655 | 0.95 | 0.09, 9.87 | Increased | 0.97 | >1 | 0.15 | 51 | F |
| *Activating KIR genes* | | | | | | | | | | | | |
| *2DS1* | All | 6 | 260/834 | 211/758 | 0.95 | 0.75, 1.20 | Decreased | 0.68 | >1 | 0.48 | 0 | F |
| | Caucasians | 3 | 267/643 | 88/244 | 1.27 | 0.69, 2.33 | Increased | 0.44 | >1 | 0.04 | 68 | R |
| | Caucasians* | 2 | 223/520 | 46/147 | **1.71** | **1.15, 2.53** | **Increased** | **0.007** | 0.49 | 0.90 | 0 | F |
| | Asians | 2 | 120/256 | 79/151 | 0.90 | 0.59, 1.35 | Decreased | 0.60 | >1 | 0.84 | 0 | F |
| | Africans | 2 | 64/394 | 79/481 | 1.04 | 0.66, 1.63 | Increased | 0.88 | >1 | 0.23 | 30 | F |
| *2DS2* | All | 6 | 428/834 | 386/758 | 1.08 | 0.88, 1.32 | Increased | 0.48 | >1 | 0.68 | 0 | F |
| | Caucasians | 3 | 355/643 | 112/244 | 1.42 | 0.97, 2.08 | Increased | 0.07 | >1 | 0.25 | 28 | F |
| | Asians | 2 | 120/256 | 75/151 | 1.27 | 0.59, 2.75 | Increased | 0.55 | >1 | 0.15 | 51 | F |
| | Africans | 2 | 211/394 | 249/481 | 1.04 | 0.80, 1.36 | Increased | 0.77 | >1 | 0.87 | 0 | F |
| *2DS3* | All | 6 | 346/1,084 | 216/772 | 1.22 | 0.87, 1.73 | Increased | 0.25 | >1 | 0.06 | 53 | R |
| | All* | 4 | 168/564 | 186/625 | 0.97 | 0.75, 1.25 | Null | 0.81 | >1 | 0.48 | 0 | F |
| | Caucasians | 3 | 213/643 | 59/244 | 1.53 | 0.87, 2.17 | Increased | 0.14 | >1 | 0.11 | 55 | F |
| | Africans | 2 | 103/394 | 133/481 | 0.91 | 0.67, 1.23 | Decreased | 0.53 | >1 | 0.47 | 0 | F |
| *2DS4F* | All | 3 | 268/348 | 302/402 | **1.62** | **1.10, 2.37** | **Increased** | **0.01** | 0.70 | 0.15 | 47 | F |
| | Caucasians | 3 | 243/643 | 75/244 | 1.47 | 0.64, 3.41 | Increased | 0.37 | >1 | 0.004 | 82 | R |
| | Caucasians* | 2 | 182/520 | 52/147 | 0.97 | 0.66, 1.42 | Null | 0.87 | >1 | 0.93 | 0 | F |
| | Africans | 1 | 209/240 | 251/326 | **2.01** | **2.01, 3.18** | **Increased** | **0.003** | 0.21 | NA | NA | NA |
| *2DS4D* | All | 5 | 688/930 | 425/617 | 0.86 | 0.67, 1.10 | Decreased | 0.24 | >1 | 1.00 | 0 | F |

(*Continued*)

**Table 2.** (Continued)

| *KIR* | Ethnicity | K | HIVI (n/N) | HESN (n/N) | Test of association | | | | | Test of heterogeneity | | AM |
|---|---|---|---|---|---|---|---|---|---|---|---|---|
| | | | | | OR | 95% CI | Risk | Pᵃ | Pᶜ | Pᵇ | I² (%) | |
| | Caucasians | 3 | 522/643 | 199/244 | 0.88 | 0.59, 1.30 | Decreased | 0.51 | >1 | 0.97 | 0 | F |
| | Africans | 1 | 128/240 | 188/326 | 0.84 | 0.60, 1.17 | Decreased | 0.30 | >1 | NA | NA | NA |
| *2DS5* | All | 6 | 400/1,084 | 351/772 | 0.89 | 0.73, 1.09 | Decreased | 0.27 | >1 | 0.67 | 0 | F |
| | Caucasians | 3 | 184/643 | 74/244 | 0.97 | 0.65, 1.45 | Null | 0.89 | >1 | 0.27 | 24 | F |
| | Africans | 2 | 181/394 | 241/481 | 0.83 | 0.64, 1.09 | Decreased | 0.19 | >1 | 0.94 | 0 | F |
| *3DS1* | All | 5 | 186/773 | 172/729 | **0.76** | **0.57, 1.00** | **Decreased** | **0.05** | >1 | 0.17 | 37 | F |
| | Caucasians | 3 | 246/643 | 89/244 | 1.12 | 0.71, 1.76 | Increased | 0.64 | >1 | 0.17 | 44 | F |
| | Africans | 3 | 41/491 | 55/655 | 1.19 | 0.56, 2.56 | Increased | 0.65 | >1 | 0.07 | 63 | R |
| | Africans* | 2 | 22/251 | 17/329 | 1.80 | 0.93, 3.48 | Increased | 0.08 | >1 | 0.65 | 0 | F |

K: number of studies; HIV: Human Immunodeficiency Virus; HIVI: HIV-Infected; HESN: *HIV-exposed seronegative*; n: number of individuals; N: total number; OR: odds ratio; CI: confidence interval; Null: OR 0.97–1.03; Pᵃ: P-value for test of association; Pᶜ: Bonferroni corrected Pᵃ; Pᵇ: P-value for heterogeneity; I² is a measure of variability; Values in **bold** indicate significant associations; F: Fixed-effects; R: Random-effects; AM: Analysis Model; NA: Not applicable;* outlier treated

treatments. Between the two polymorphisms, *2DL3* presents strong evidence on account of the magnitude of protective effect (81%), associative and interaction outcomes ($P^{ci} < 10^{-4}$). On the other hand, *3DS1S1* is strong based on number of studies and aggregate statistical power (Table 6). The advantage or disadvantages of using sensitivity approach versus eliminating the outlier is contextualized in terms of the following: Sensitivity treatment evaluates robustness of the pooled ORs while outlier elimination addresses heterogeneity. Favorable outcome of sensitivity analysis is robustness, where no study contributed to instability of the results. On the other hand, favorable outcomes of outlier treatment involve both heterogeneity and significance. In our study, heterogeneity was either reduced or eliminated; significance was

**Table 3. Summary associations of *3DL1/S1* genotypes and HIV acquisition in the pre-outlier (PRO) analysis.**

| *KIR* genotype | Comparisons | PRO | | | | | | | | | | | AM |
|---|---|---|---|---|---|---|---|---|---|---|---|---|---|
| | | K | HIVI (n/N) | HESN (n/N) | Test of association | | | | | Test of heterogeneity | | | |
| | | | | | OR | 95%CI | Risk | Pᵃ | Pᶜ | Pᵇ | I² (%) | | |
| *3DL1L1* | All | 10 | 1,181/1,850 | 456/717 | 1.19 | 0.83, 1.71 | Increased | 0.34 | >1 | 0.01 | 60 | | R |
| | Caucasians | 7 | 1,010/1,629 | 286/494 | 1.20 | 0.81, 1.77 | Increased | 0.36 | >1 | 0.01 | 65 | | R |
| | Asians | 1 | 13/47 | 5/47 | **3.21** | **1.04, 9.90** | **Increased** | **0.04** | >1 | NA | NA | | NA |
| | Africans | 2 | 158/174 | 165/176 | 0.67 | 0.30, 1.49 | Decreased | 0.33 | >1 | 0.45 | 0 | | F |
| *3DL1S1* | All | 10 | 574/1,850 | 201/717 | 1.01 | 0.73, 1.41 | Null | 0.94 | >1 | 0.03 | 52 | | R |
| | Caucasians | 7 | 535/1,629 | 157/494 | 1.07 | 0.77, 1.47 | Increased | 0.70 | >1 | 0.09 | 45 | | R |
| | Asians | 1 | 24/47 | 35/47 | **0.36** | **0.15, 0.85** | **Decreased** | **0.02** | >1 | NA | NA | | NA |
| | Africans | 2 | 15/174 | 9/176 | 1.73 | 0.73, 4.09 | Increased | 0.21 | >1 | 0.52 | 0 | | F |
| *3DS1S1* | All | 8 | 94/1,850 | 58/717 | 0.54 | 0.29, 1.01 | Decreased | 0.06 | >1 | 0.02 | 59 | | R |
| | Caucasians | 7 | 84/1,629 | 51/494 | **0.45** | **0.24, 0.84** | **Decreased** | **0.01** | 0.70 | 0.06 | 50 | | R |
| | Asians | 1 | 10/47 | 7/47 | 1.54 | 0.53, 4.48 | Increased | 0.42 | >1 | NA | NA | | NA |

PRO: pre-outlier; K: number of studies; HIV: Human Immunodeficiency Virus; HIVI: HIV-Infected; HESN: *HIV-exposed seronegative*; n: number of individuals; N: total number; OR: odds ratio; CI: confidence interval; Null: OR 0.97–1.03; Pᵃ: P-value for test of association; Pᶜ: Bonferroni corrected Pᵃ; Pᵇ: P-value for heterogeneity; I² is a measure of variability; F: Fixed-effects; R: Random-effects; AM: Analysis Model; NA: Not applicable; Values in **bold** indicate significant associations.

**Table 4. Summary associations of *3DL1/S1* genotypes and HIV acquisition in the post-outlier (PSO) analysis.**

| *KIR* genotype | Ethnicity | K | HIVI (n/N) | HESN (n/N) | PSO | | | | | | | AM | Effects of outlier treatment |
| | | | | | Test of association | | | | | Test of heterogeneity | | | |
| | | | | | OR | 95%CI | Risk | $P^a$ | $P^c$ | $P^b$ | $I^2$ (%) | | |
| *3DL1L1* | All | 6 | 796/1,183 | 349/500 | 1.19 | 0.92, 1.53 | Increased | 0.18 | >1 | 0.47 | 0 | F | EH, NC |
| | Caucasians | 4 | 638/1,009 | 184/324 | 1.27 | 0.98, 1.66 | Increased | 0.08 | >1 | 0.62 | 0 | F | EH, NC |
| | Africans | 2 | 158/174 | 165/176 | 0.66 | 0.30, 1.46 | Decreased | 0.30 | >1 | 0.45 | 0 | F | NC, NC |
| *3DL1S1* | All | 8 | 508/1,703 | 149/647 | 1.21 | 0.97, 1.51 | Increased | 0.10 | >1 | 0.8 | 0 | F | EH, NC |
| | Caucasians | 6 | 493/1,529 | 140/471 | 1.17 | 0.93, 1.48 | Increased | 0.18 | >1 | 0.75 | 0 | F | EH, NC |
| | Africans | 2 | 15/174 | 9/176 | 1.75 | 0.75, 4.12 | Increased | 0.20 | >1 | 0.52 | 0 | F | NC, NC |
| *3DS1S1* | Caucasians | 6 | 62/1170 | 48/376 | **0.37** | **0.24, 0.56** | **Decreased** | $< 10^{-5}$ | $< 10^{-3}$ | 0.38 | 5 | F | RH, IS |

PSO: post-outlier; K: number of studies; HIV: Human Immunodeficiency Virus; HIVI: HIV-Infected; HESN: *HIV-exposed seronegative*; n: number of individuals; N: total number; OR: odds ratio; CI: confidence interval; $P^a$: P-value for test of association; $P^c$: Bonferroni correction for $P^a$; $P^b$: P-value for heterogeneity; $I^2$ is a measure of variability; F: Fixed-effects; AM: Analysis Model; EH: eliminated heterogeneity; RH: reduced heterogeneity; IS: intensified significance; NC: no change; Values in **bold** indicate significant associations

intensified. These effects from outlier treatment and those from sensitivity analysis, contribute to strengthening the evidence that we present.

## Functional correlates

Between our two main findings, *3DS1S1* appears to have stronger support from functional studies than *2DL3*. Because *3DS1* is more prominent in the HIV literature [48] than *2DL3*, functional correlate narrative here refer to *3DS1*. In the proposed model explaining results based on the concept of "NK licensing", individuals carrying *3DS1* would lead to stronger NK cell activation by degranulation and cytokine release to control early HIV-1 infection [49, 50]. Essentially, functional studies support the protective effect of *3DS1* [51–53]. An increase IFN-γ and CD107a expressions of NK cells were observed in *3DS1* individuals with early HIV-1 infection [52].

The roles of 3DS1+NK cells in HIV infection are two-fold, one, is expansion in acute HIVI individuals [15] and the other is increased antiviral activity in HIV-infected cells [49]. The nature of *KIR* influence on HIV-infection is admittedly more complex than the sum of the meta-analytical evidence and functional support for our findings. The complexity is made

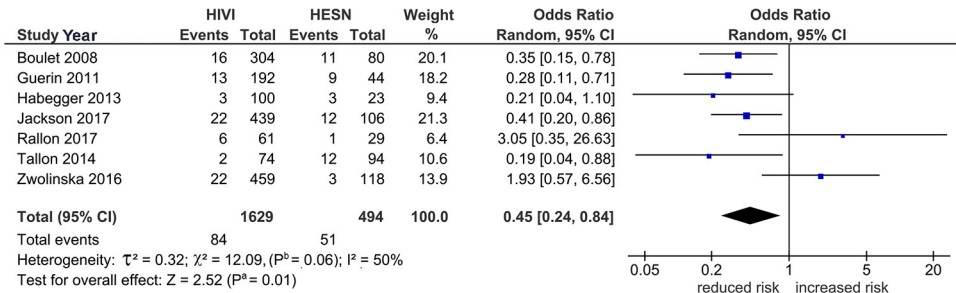

**Fig 2. Pre-outlier (PRO) summary effects of *3DS1S1* on HIV acquisition in Caucasians.** Diamond denotes the pooled odds ratio (OR) indicating reduced risk (OR 0.45). Squares show the OR of each study. Horizontal lines on either side of each square represent 95% confidence intervals (CIs). Significance from the Z test for overall effect is moderate ($P^a = 0.01$). The χ² test shows the presence of heterogeneity ($P^b = 0.06$, $I^2 = 50\%$); $I^2$: a measure of variability expressed in %.

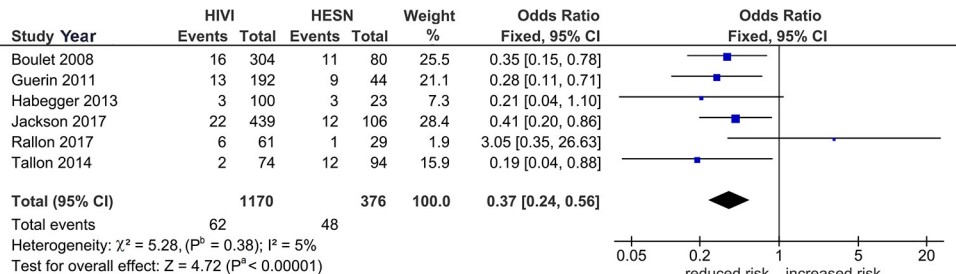

**Fig 3. Galbraith plot analysis to detect the source of heterogeneity among Caucasian studies; the study above the +2 confidence limit is the outlier, Zwolinska et al [44]; whose presence in the PRO forest plot (Fig 2) accounts for 50% of the heterogeneity.** Removal of this study [44] from the PSO forest plot (Fig 4) reduced the heterogeneity to 5%. OR: odds ratio; SE: standard error.

| | HIVI | | HESN | | Weight | Odds Ratio | | Odds Ratio |
|---|---|---|---|---|---|---|---|---|
| Study Year | Events | Total | Events | Total | % | Fixed, 95% CI | | Fixed, 95% CI |
| Boulet 2008 | 16 | 304 | 11 | 80 | 25.5 | 0.35 [0.15, 0.78] | | |
| Guerin 2011 | 13 | 192 | 9 | 44 | 21.1 | 0.28 [0.11, 0.71] | | |
| Habegger 2013 | 3 | 100 | 3 | 23 | 7.3 | 0.21 [0.04, 1.10] | | |
| Jackson 2017 | 22 | 439 | 12 | 106 | 28.4 | 0.41 [0.20, 0.86] | | |
| Rallon 2017 | 6 | 61 | 1 | 29 | 1.9 | 3.05 [0.35, 26.63] | | |
| Tallon 2014 | 2 | 74 | 12 | 94 | 15.9 | 0.19 [0.04, 0.88] | | |
| | | | | | | | | |
| Total (95% CI) | | 1170 | | 376 | 100.0 | 0.37 [0.24, 0.56] | | |
| Total events | 62 | | 48 | | | | | |

Heterogeneity: $\chi^2 = 5.28$, ($P^b = 0.38$); $I^2 = 5\%$
Test for overall effect: $Z = 4.72$ ($P^a < 0.00001$)

reduced risk   increased risk

**Fig 4. Post-outlier (PSO) summary effects of *3DS1S1* on HIV acquisition in Caucasians.** Diamond denotes the pooled odds ratio (OR) indicating reduced risk (OR 0.37). Squares show the OR of each study. Horizontal lines on either side of each square represent 95% confidence intervals (CIs). Significance from the Z test for overall effect is high ($P^a < 0.00001$). The $\chi^2$ test shows reduced heterogeneity ($P^b = 0.38$, $I^2 = 5\%$); $I^2$: a measure of variability expressed in %.

**Table 5. Sensitivity analysis outcomes.**

| *KIR* genes content | | | |
|---|---|---|---|
| polymorphism | Population | Genetic effects | |
| *2DL1* | Caucasians | Robust | |
| *2DL2* | Caucasians | [44, 46] | |
| *2DL3* | Caucasians | Robust | |
| *2DS1* | Caucasians | [44] | |
| *2DS4F* | All | Robust | |
| *3DS1* | All | [35, 38, 45] | |
| ***3DL1/S1* genotype** | | | |
| polymorphism | | PRO | PSO |
| *3DS1S1* | All | None | Robust |
| *3DS1S1* | Caucasians | [40, 41, 45] | Robust |

PRO: pre–outlier; PSO: post-outlier; the value in brackets indicate the reference articles that contributed to instability of associations.

more elaborate from three viewpoints: (i) *in vivo/in vitro* effects of *KIR* on HIVI; (ii) extensive genetic diversity of *KIR* among populations; and (iii) influence of linkage disequilibrium, raising the possibility that the observed effect maybe mediated by *3DS1* or other *KIRs*.

## *KIR* polymorphisms in meta-analysis

To our knowledge, this is the first meta-analysis that examines *KIR* effects on HIV acquisition. By extension, associations of the *KIR* polymorphisms have been reported in a number of meta-analyses that included disease endpoints such as systemic lupus erythematosus, rheumatoid arthritis, type 1 diabetes mellitus and multiple sclerosis [54–57]. The only other meta-analysis for *KIR* polymorphisms with another infectious disease is that of Gauthiez et al's

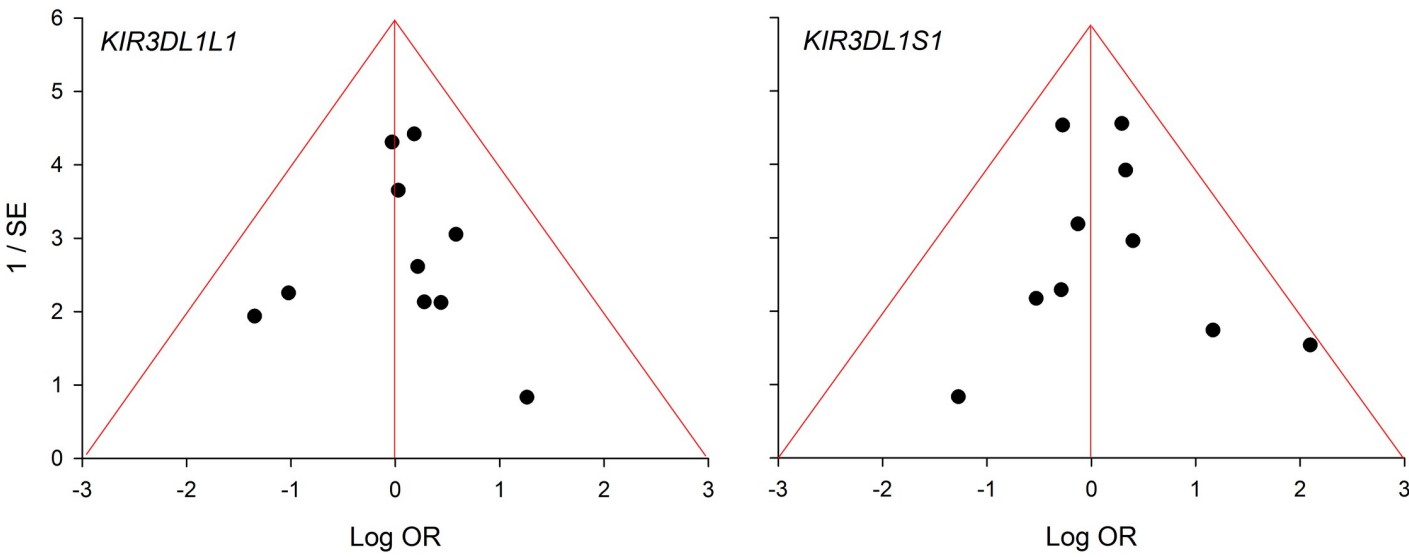

**Fig 5. Funnel plot analysis of *3DL1/S1* genotype for publication bias.** OR: odds ratio; SE: standard error.

**Table 6. Comparative summary effects between *2DL3* and *3DS1S1* on HIV acquisition in Caucasians in PSO.**

| Parameter | 2DL3 | 3DS1S1 |
|---|---|---|
| N | 2 | 6 |
| n | 677 | 1,546 |
| Aggregate statistical power | 57% | 92% |
| OR | 0.19 | 0.37 |
| Magnitude of protective effect | 81% | 63% |
| 95% CI | 0.09, 0.40 | 0.24, 0.56 |
| CI difference (upper CI-lower CI) | 0.31 | 0.32 |
| Direction of risk effects | Decreased | Decreased |
| P$^c$ | $< 10^{-3}$ | $< 10^{-3}$ |
| P$^{ci}$ | $< 10^{-4}$ | 0.14 |
| Sensitivity analysis outcomes | Robust | Robust |

PSO: post-outlier; N: number of included studies; n: sample size; P$^c$: Bonferroni-corrected P-value; P$^{ci}$: Bonferroni-corrected P-value for interaction; OR: odd ratio; CI: confidence interval

examination of the Hepatitis C Virus (HCV) infection with HCV clearance [58]. Owing to the incompatibility of results, we compare the two meta-analyses based on methodology. S6 Table summarizes the comparative features of the two meta-analyses. In common between the two studies are the uses of I$^2$ to evaluate heterogeneity and Mantel-Haenszel and DerSimonian-Laird for fixed and random-effects, respectively. Meta-analysis features covered in this study but not in Gauthiez et al [58] were assessment of study quality, interaction test, outlier treatment and correction for multiplicity.

## *KIR* and GWAS

Genome-wide association studies (GWAS) is a powerful approach to unravel the genetics behind complex diseases [59]. In HIV research, GWAS has identified a number of SNPs associated with different forms of HIV progression [60]. The first GWAS in the HIV context was in the HLA class I locus that confirmed a major effect of *HLA-B*57* in reducing viral load [61]. Containment of viral load in the early stages of HIV infection is facilitated by the HLA-B/KIR genotype which enhances activation of NK cells [62]. Evidence for KIR-HLA suggests complex interactions but GWAS appears to be problematic in examining the role of this locus in the genome context [63]. The reason for this problematic approach relates to the following: One, HLA-KIR molecules are encoded by two of the most diverse gene families in the human genome [64]. Diversity of the HLA and KIR loci impacts viral pathogenesis differentially across individuals [64]. Two, the *KIR* locus contains variations of the KIR genes. This variation is functionally relevant only in the presence of alleles encoding their specific HLA ligands [63]. For example, disabled protectivity of the HLA-B allele without *3DS1* contrasts with *3DS1*-related AIDS progression in the absence of specific HLA-B alleles [65]. Thus, variation in the genes encoding KIR proteins, particularly *3DL1* and *3DS1*, has been associated with HIV-1 outcomes in many genetic and functional studies [66], but these have not been identified by GWAS, almost certainly because of the extreme inter- and intragenic variability of the *KIR* haplotypes [67]. Three, on the fundamental level, the agnostic approach of GWAS in analyzing SNPs limits the assessment of functionally dependent variants such as that shown by HLA-KIR [63].

## Strengths and limitations

Our results are better contextualized with awareness of their strengths and limitations. The strengths include: (i) impact of outlier treatment on associative significance and heterogeneity;

(ii) added evidence of the high methodological quality of all 13 articles with CBS scores of $\geq 5$; (iii) of the 70 comparisons, 53 (81%) were non-heterogeneous (fixed-effects); of the 53, 31 (58%) had zero heterogeneity ($I^2 = 0\%$); (v) one core finding (*3DS1S1*) in the genotype analysis had high statistical power (92%); (vi) sensitivity treatment confirmed robustness of our core findings. On the other hand, limitations comprise of the following: (i) effects of gene-gene and gene-environment interactions were not addressed due to the lack of adequate data; (ii) few studies for Africans and Asians resulted in under-representation of these ethnic groups; (iii) the linkage disequilibrium effect may involve other proximal *KIR* polymorphisms that might account for the associations; (iv) 10 comparisons had only one study (four Asians, four Africans and two Caucasians) and (v) one core finding (*2DL3*) in the gene content analysis were statistically underpowered (57%).

## Conclusion

This study hopes to contribute to the genetic knowledge of this epidemiologically important infectious disease. Although our findings are admittedly modest, they profile the role of the two polymorphisms (*2DL3* and *3DS1S1*) in HIV acquisition. Considered individually, other *KIR* polymorphisms may have influence and would probably require analyses of haplotypes and HLA ligands to distinguish combined effects. These approaches may elaborate on how genetic variation cooperates in NK-mediated protection against HIV infection. Such analyses may shed light on the complexities of *KIR*'s involvement in the innate immune responses of HIV acquisition.

## Supporting information

**S1 List. Excluded studies after abstract screening and full-text articles assessed for eligibility.**
(DOCX)

**S1 Table. Characteristics of the studies in the *KIR* gene content polymorphisms and its associations with HIV acquisition.**
(DOCX)

**S2 Table. Characteristics of the studies in the *3DL1/S1* genotype polymorphisms and its associations with HIV acquisition.**
(DOCX)

**S3 Table. PRISMA checklist.**
(DOCX)

**S4 Table. Genetic association checklist.**
(DOCX)

**S5 Table. Tests of interaction.**
(DOCX)

**S6 Table. Comparison of two meta-analyses based on methodology.**
(DOCX)

## Acknowledgments

SC was supported by Thailand Research Fund and the Commission on Higher Education under the Grant for New Researcher (No.MRG 6180172).

## Author Contributions

**Conceptualization:** Suwit Chaisri, Chanvit Leelayuwat.

**Data curation:** Suwit Chaisri, Phuntila Tharabenjasin.

**Formal analysis:** Suwit Chaisri, Noel Pabalan, Sompong Tabunhan, Phuntila Tharabenjasin.

**Funding acquisition:** Suwit Chaisri.

**Investigation:** Suwit Chaisri, Sompong Tabunhan, Nipaporn Sankuntaw, Chanvit Leelayuwat.

**Methodology:** Suwit Chaisri, Noel Pabalan, Phuntila Tharabenjasin.

**Project administration:** Suwit Chaisri.

**Resources:** Suwit Chaisri.

**Software:** Suwit Chaisri, Noel Pabalan.

**Supervision:** Suwit Chaisri, Noel Pabalan, Chanvit Leelayuwat.

**Validation:** Suwit Chaisri, Noel Pabalan, Sompong Tabunhan, Chanvit Leelayuwat.

**Visualization:** Suwit Chaisri, Noel Pabalan.

**Writing – original draft:** Suwit Chaisri, Noel Pabalan.

**Writing – review & editing:** Suwit Chaisri, Noel Pabalan, Sompong Tabunhan, Phuntila Tharabenjasin, Nipaporn Sankuntaw, Chanvit Leelayuwat.

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
