## [Decision Letter · Decision Letter 0]

7 Aug 2019

PONE-D-19-15826

Effects of the killer immunoglobulin–like receptor (KIR) polymorphisms on HIV acquisition: a meta-analysis

PLOS ONE

Dear Dr. Pabalan,

Thank you for submitting your manuscript to PLOS ONE. Both reviewers felt that the study was well designed and executed. There are some minor concerns that were expressed which could be easily addressed.

We would appreciate receiving your revised manuscript by 10/1/2019. To enhance the reproducibility of your results, we recommend that if applicable you deposit your laboratory protocols in protocols.io, where a protocol can be assigned its own identifier (DOI) such that it can be cited independently in the future. For instructions see: http://journals.plos.org/plosone/s/submission-guidelines#loc-laboratory-protocols

We look forward to receiving your revised manuscript.

Kind regards,

Srinivas Mummidi, D.V.M., Ph.D.

Academic Editor

PLOS ONE

Journal Requirements:

 [SC was supported by Thailand Research Fund and the Commission on Higher Education under the Grant for New Researcher (No.MRG 6180172)]. 

Reviewers' comments:

Reviewer's Responses to Questions

**Comments to the Author**

1. Is the manuscript technically sound, and do the data support the conclusions?

Reviewer #1: Yes

Reviewer #2: Yes

2. Has the statistical analysis been performed appropriately and rigorously? 

Reviewer #1: Yes

Reviewer #2: Yes

3. Have the authors made all data underlying the findings in their manuscript fully available?

Reviewer #1: Yes

Reviewer #2: Yes

4. Is the manuscript presented in an intelligible fashion and written in standard English?

Reviewer #1: Yes

Reviewer #2: Yes

5. Review Comments to the Author

Reviewer #1: The authors have done a terrific job at reviewing the relevant literature on the role of KIR polymorphisms in HIV acquisition, and, more importantly, at extracting the data and meta-data from these studies to perform a meta-analysis. The meta-analysis was satisfactorily performed. Scoring and ranking studies according to the CB scale lends credence to the meta-analysis to the extent that it establishes the quality of the "input" studies.

A major finding of the study is that the 2DL3 polymorphism is significantly protective against HIV acquisition even after Bonferroni correction. This was established only after outlier removal where outlier studies were identified by heterogeneity analysis and removed accordingly. The subsequent post-outlier treatment analysis yielded a strong inference that 2DL3 is indeed protective against HIV acquisition.

There are two very minor suggestions that may strengthen the paper a little bit:

1) Clarification and interpretation of the significant interaction result between Caucasians and Africans. What does this mean even simply at an epidemiological level?

2) The Benjamini-Hochberg false discovery rate (BH FDR) is often a desirable alternative to the more rigorous Bonferroni correction in that it represents a nice balance between guarding against false discoveries and making true discoveries, as opposed to simply guarding against false discoveries in a relatively more conservative manner (as in the Bonferroni correction). This is not necessary for the authors to implement but its implementation may yield more results surviving multiple correction if the BH FDR is used.

Reviewer #2: This is an interesting meta-analysis for effects of genetic variants on HIV acquisition.

The paper was well written and methods were rigorously used and described. A minor comment regarding the Galbraith plot. Was it based on an specific protective size of effect? Please define in the text if you did so. It is advisable to the authors include the size of effect plot (arc) on Figure 3. Define Zwoliska in the same figure legend. This unique point was responsible for the 45% heterogeneity between studies. There was not mention on the discussion regarding the advantage or disadvantages of using sensitivity approach versus eliminating the outlier.

Table 4 should clarify if the numbers in brackets are those affecting the results? (i.e.: Modifying the ORs in extreme results?). Can you clarify their significance?

6. PLOS authors have the option to publish the peer review history of their article (what does this mean?). If published, this will include your full peer review and any attached files.

Reviewer #1: No

Reviewer #2: No

---

## [Author Response · Author response to Decision Letter 0]

31 Aug 2019

Reviewer #1: The authors have done a terrific job at reviewing the relevant literature on the role of KIR polymorphisms in HIV acquisition, and, more importantly, at extracting the data and meta-data from these studies to perform a meta-analysis. The meta-analysis was satisfactorily performed. Scoring and ranking studies according to the CB scale lends credence to the meta-analysis to the extent that it establishes the quality of the "input" studies.

A major finding of the study is that the 2DL3 polymorphism is significantly protective against HIV acquisition even after Bonferroni correction. This was established only after outlier removal where outlier studies were identified by heterogeneity analysis and removed accordingly. The subsequent post-outlier treatment analysis yielded a strong inference that 2DL3 is indeed protective against HIV acquisition.

There are two very minor suggestions that may strengthen the paper a little bit:

COMMENT:

1) Clarification and interpretation of the significant interaction result between Caucasians and Africans. What does this mean even simply at an epidemiological level?

RESPONSE: 

We clarify and interpret the significant interaction result between Caucasians and Africans. Prevalence of HIV in Africa is higher compared to that in North America and Europe (Caucasians) [1]. This epidemiological finding suggests less risk for Caucasians and higher risk for Africans which agrees with our 2DL3 result that protects Caucasians (OR, 0.19) from risk of HIV acquisition. 

LINES 306-308

Extent of the significant Caucasian effect is thus placed in context when compared with its non-significant African counterpart. 

COMMENT:

2) The Benjamini-Hochberg false discovery rate (BH FDR) is often a desirable alternative to the more rigorous Bonferroni correction in that it represents a nice balance between guarding against false discoveries and making true discoveries, as opposed to simply guarding against false discoveries in a relatively more conservative manner (as in the Bonferroni correction). This is not necessary for the authors to implement but its implementation may yield more results surviving multiple correction if the BH FDR is used.

RESPONSE: 

We sincerely thank the reviewer for the insight provided about correcting for multiple comparisons. We did implement BH-FDR and obtained 11 significant P-values against the two with the Bonferroni correction. Then we found that explaining the 11 significant outcomes did not readily converge with the epidemiology and physiology literature. From the BH-FDR approach, this made the explanations messy which muddled the principal message of our study. Thus, we found that applying the Bonferroni correction, conservative as it is, readily lent more credence to our main message with better support from the literature.

*The Table at the end of this document puts the BH-FDR results in column 

Reviewer #2: This is an interesting meta-analysis for effects of genetic variants on HIV acquisition.

The paper was well written and methods were rigorously used and described. 

COMMENT:

A minor comment regarding the Galbraith plot. Was it based on an specific protective size of effect? 

RESPONSE: 

We thank the reviewer for this question. The Galbraith plot is mainly used to find source(s) of heterogeneity in meta-analysis. Thus, it would identify which component study/studies are outliers. In calculating the values along the Y-axis (log OR / SE), we indeed use the range of effect sizes (which maybe from protective to increased risk) from all the included studies in order to derive the log values. Rather than the specific protective size of effect as basis, it is the consequence of outlier analysis, which the Galbraith plot is the main instrument in identifying the outlier(s).

COMMENT:

Please define in the text if you did so. It is advisable to the authors include the size of effect plot (arc) on Figure 3. 

RESPONSE: 

The reason we use the log OR /SE is to derive positive and negative confidence limits for the Galbraith plot along the y-axis. Unfortunately, size of effect plot (arc) did not allow the above when we attempted to do so.

COMMENT:

Define Zwoliska in the same figure legend. This unique point was responsible for the 45% heterogeneity between studies. 

RESPONSE: In the legend of figure 3 (below) we define the role of the outlying study, Zwolinska et al in terms of the heterogeneity in the Caucasian forest plot of the 3DS1S1 genotype comparison.

LINES 621-625

Figure 3: Galbraith plot analysis to detect the source of heterogeneity among Caucasian studies; the study above the +2 confidence limit is the outlier, Zwolinska et al [44]; whose presence in the PRO forest plot (Figure 2) accounts for 50% of the heterogeneity. Removal of this study [44] from the PSO forest plot (Figure 4) reduced the heterogeneity to 5%. OR: odds ratio; SE: standard error.

COMMENT:

There was not mention on the discussion regarding the advantage or disadvantages of using sensitivity approach versus eliminating the outlier.

RESPONSE: We thank the reviewer for this comment. We explain the pros and cons of using sensitivity analysis in the Discussion. Mainly, pooled ORs need to be tested for their stability. We address the question: will any study-specific OR be responsible for the instability of the pooled OR? This is tested simply by omitting one study at a time then recalculating the pooled OR without that study. 

LINES 356-363

The advantage or disadvantages of using sensitivity approach versus eliminating the outlier is contextualized in terms of the following: Sensitivity treatment evaluates robustness of the pooled ORs while outlier elimination addresses heterogeneity. Favorable outcome of sensitivity analysis is robustness, where no study contributed to instability of the results. On the other hand, favorable outcomes of outlier treatment involve both heterogeneity and significance. In our study, heterogeneity was either reduced or eliminated; significance was intensified. These effects from outlier treatment and those from sensitivity analysis, contribute to strengthening the evidence that we present.

COMMENT:

Table 4 should clarify if the numbers in brackets are those affecting the results? (i.e.: Modifying the ORs in extreme results?). Can you clarify their significance?

RESPONSE: The bracketed numbers in the sensitivity analysis Table 4 are simply indications of the references that rendered non-robustness to the comparisons. Removing a study that results in a change of the pooled OR from significant (P < 0.05) to non-significant (P > 0.05) indicates non-robustness (instability) of the comparisons. The significance of the number brackets (Table 4) is a measure of stability/instability of the pooled results.

LINES 336-337

PRO: pre–outlier; PSO: post-outlier; the value in brackets indicate the references articles that contributed to instability of associations.

References

1. Maartens G, Celum C, Lewin SR (2014) HIV infection: epidemiology, pathogenesis, treatment, and prevention. Lancet 384: 258-271.

---

## [Decision Letter · Decision Letter 1]

30 Oct 2019

Effects of the killer immunoglobulin–like receptor (KIR) polymorphisms on HIV acquisition: a meta-analysis

PONE-D-19-15826R1

Dear Dr. Pabalan,

We are pleased to inform you that your manuscript has been judged scientifically suitable for publication and will be formally accepted for publication once it complies with all outstanding technical requirements.

With kind regards,

Srinivas Mummidi, D.V.M., Ph.D.

Academic Editor

PLOS ONE

Additional Editor Comments (optional):

Reviewers' comments:

Reviewer's Responses to Questions

**Comments to the Author**

1. If the authors have adequately addressed your comments raised in a previous round of review and you feel that this manuscript is now acceptable for publication, you may indicate that here to bypass the “Comments to the Author” section, enter your conflict of interest statement in the “Confidential to Editor” section, and submit your "Accept" recommendation.

Reviewer #1: All comments have been addressed

Reviewer #2: All comments have been addressed

2. Is the manuscript technically sound, and do the data support the conclusions?

Reviewer #1: Yes

Reviewer #2: Yes

3. Has the statistical analysis been performed appropriately and rigorously? 

Reviewer #1: Yes

Reviewer #2: Yes

4. Have the authors made all data underlying the findings in their manuscript fully available?

Reviewer #1: Yes

Reviewer #2: Yes

5. Is the manuscript presented in an intelligible fashion and written in standard English?

Reviewer #1: Yes

Reviewer #2: Yes

6. Review Comments to the Author

Reviewer #1: The authors have satisfactorily addressed my comments. The authors are to be congratulated for a fine manuscript worthy of publication in the journal.

Reviewer #2: The authors made the suggested changes in the manuscript. The paper is well designed and the statistics well performed. The sections of the paper accomplish the PRISMA requirements.

7. PLOS authors have the option to publish the peer review history of their article (what does this mean?). If published, this will include your full peer review and any attached files.

Reviewer #1: No

Reviewer #2: No

---

## [Editor Report · Acceptance letter]

8 Nov 2019

PONE-D-19-15826R1 

Effects of the killer immunoglobulin–like receptor (*KIR*) polymorphisms on HIV acquisition: a meta-analysis 

Dear Dr. Pabalan:

I am pleased to inform you that your manuscript has been deemed suitable for publication in PLOS ONE. Congratulations! Your manuscript is now with our production department. 

With kind regards,

on behalf of

Dr. Srinivas Mummidi 

Academic Editor

PLOS ONE